# Metamorphic Adversarial Detection Pipeline for Face Recognition Systems

**Rohan Reddy Mekala,**[1] **Sai Yerramreddy,** [2] **Adam Porter** [2]

[1] Fraunhofer USA CESE, UMD Riverdale, Maryland, USA
[2] Department of Computer Science, University of Maryland, College Park, USA
rreddy@fc-md.umd.edu, saiyr@umd.edu, aporter@umd.edu

## Abstract

Adversarial examples pose a serious threat to the robustness of machine learning models in general and deep learning models in particular. Computer vision tasks like image classification, facial recognition, object detection, etc. and natural language processing tasks like sentiment analysis and semantic similarity assessment have all been proven vulnerable to adversarial attacks. For computer vision tasks specifically, these carefully crafted perturbations to input images can cause targeted misclassifications to a label of the attacker's choice, without the perturbations being detectable to the naked eye. A particular class of adversarial attacks called black box attacks can be used to fool a model under attack despite not having access to the model parameters or input datasets used to train the model. As part of the research presented in this paper, we first deploy a range of state of the art adversarial attacks against multiple face recognition pipelines trained in a black box setup, and then generate pair-wise adversarial image sets to deceive the corresponding models under attack. Consequently, we propose a novel approach for adversarial detection that utilizes statistical techniques to learn optimal thresholds of separation between clean and adversarial examples; achieving state of the art detection accuracies of over 90%. Our proposed method has been exhaustively tested on multiple face recognition models under attack and adversarial attack type combinations with encouraging results.

## Introduction

Deep learning has made incredible progress in recent years (Ambroggi 2014), forming the backbone of consumer facing software systems for self-driving cars, malware and intrusion detection systems, etc. These software systems incorporating deep learning architectures have been deployed in production due to their ability to model complex data distributions and drastically improve on traditional computer vision algorithms in generalizing over image recognition tasks like image classification, object detection, segmentation, etc. Facial recognition systems are another example of sub-tasks within the computer vision domain that have exhibited state of the art results by utilizing a deep learning driven approach to identity classification. These deep-learning backed facial recognition systems (Dmello et al. 2019) have become an important component of many industries requiring a direct interface to the customer across domains like security, retail, marketing, health-care, etc. As positive advances in deploying these facial recognition systems for real-time usage are being made, the robustness and security of the underlying deep learning methods have been called into question. For instance, it has been proven that these deep learning systems can be fooled (I. Goodfellow 2014; C. Szegedy 2013) by perturbing the input images with the slightest of margins. While the discernible human eye observes no difference between the original and perturbed image, the *attack* (added perturbation) causes the output label from the model to be drastically different for the clean and perturbed image pairs. We can see in Figure 1 that adding a small amount of carefully engineered perturbation causes the facial recognition algorithm to mis-classify Hugo Chavez as Kofi Annan. Simpler perturbations added by using primitive image transforms (like rotate, resize, etc.) can be accounted for by using augmented training datasets (Shorten and Khoshgoftaar 2019) however detecting carefully crafted adversarial perturbations is much harder. Szegedy et al. 2013 were the first to come up with adversarial techniques that fooled state of the art deep neural networks around image classification tasks using minor perturbations to the images. Some of the most potent and significant adversarial attack types proposed by researchers are the Fast Gradient Sign Method (I. Goodfellow 2014), Projected Gradient Descent (Aleksander Madry 2017), Universal Perturbation Attack and Carlini and Wagner attacks (Carlini and Wagner 2017b). The Carlini-Wagner adversarial attack in particular is regarded as one of the most potent attack to validate the robustness of image recognition models. It has achieved state of the art attack success results on a wide range of machine-learning based computer vision tasks and is considered one of the hardest attacks to defend against (Carlini and Wagner 2017a). These adversarial vulnerabilities open up a whole lot of security concerns that may have adverse ramifications, especially in security sensitive applications related to face recognition. Hence it is important to develop adversarial defense techniques to counteract these adversarial vulnerabilities and thereby enable use of these models without issues of trust in production. Over the years, there has been a considerable amount of research effort put into developing various adversarial defense techniques which can be broadly categorized into 2

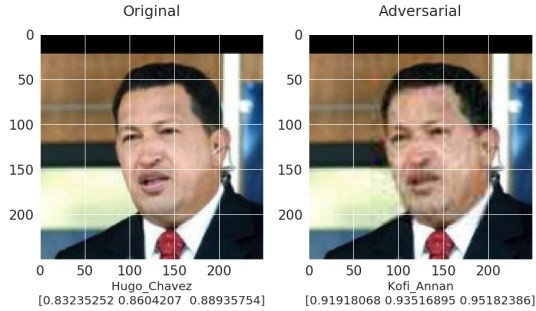

Figure 1: Adversarial attack on Hugo Chavez targeted as Kofi Annan

broad categories, 1) Adversarial Rectification, and 2) Adversarial Detection. In techniques for adversarial rectification, the corrupted input, if adversarial, is "purified" to remove the perturbation/noise component and thereby enable the model to find true output labels even for adversarial examples. On the other hand, techniques for adversarial detection work towards classifying an input as either clean or adversarial; thereby preventing adversarial versions from being input to the model. This is currently achieved by either analyzing the behavior of the model (studying the activations at the lower levels of the neural network) or by using a secondary classifier trained with real and generated adversarial samples, which can then predict if an input is adversarial or not before passing it on to the primary classifier. However, the current defense techniques proposed are mostly tested and benchmarked on low-resolution datasets like MNIST (LeCun, Cortes, and Burges 2010) and CIFAR (Krizhevsky 2009). When tried on a larger, high-resolution dataset like imagenet (Deng et al. 2009), these techniques usually don't scale well making them unsuitable for practical purposes. Additionally, these techniques suffer from the need for 1) significant compute resources/time to train the image recognition models, 2) large labeled training and test sets of adversarial examples, and 3) significant human effort and cost to label datasets. In addition, these solutions effectively hard code their defensive approach into the underlying classification model, making it difficult to adapt to new attacks over time.

In this paper, we propose and validate a novel technique for adversarial detection, which is built upon our previous research work (Rohan Reddy Mekala 2019; Mekala, Porter, and Lindvall 2020) to address all of the above-mentioned problems prevalent in current adversarial detection techniques. This proposed novel technique for adversarial defense uses metamorphic testing concepts to enable real-time detection of adversarial attacks on complex face recognition pipelines. Furthermore, the proposed methods does not require access to the output labels, attacked model's input data or parameters, is not computationally expensive and can achieve state of the art adversarial detection accuracy indifferent to the type of adversarial attack being used.

## Related Work

Image recognition is becoming a key component of many commercial and military grade systems, with obvious applications in healthcare, marketing, counter-terrorism, and other sectors. A key component of image recognition based systems, face recognition algorithms developed using deep learning have been integrated in security critical areas like payment portals, bio-metric security, and surveillance. However, the presence of adversarial vulnerabilities opens up a whole lot of security concerns that has potentially adverse consequences for businesses that adopt these "attack-unaware" deep learning models in practice. For instance, an adversarial image could cause a deep-learning based security system to grant access to an unauthorized person, leading to a potentially serious security breach. It is therefore imperative that neural network models are made robust to the threat of adversarial attacks.

### Adversarial Attacks

Adversarial examples are carefully crafted and often humanly imperceptible additive noise patterns, added to a given image. This noise reliably causes the image recognition system to output an incorrect label for the given image. A more sophisticated version of these attacks can even force the classifier to output targeted labels of the attacker's choice. Over the years, a variety of these attacks have been developed. One particular family of these adversarial attacks, called black box attacks, have been used to fool image recognition systems without the attacker needing access to the underlying image classification model, its parameters, or to its training data. For White-box attacks, the attackers have access to the gradients from the training process of the underlying model, making it comparably easier to define their attack strategy. In this paper, we evaluate our approach against 3 adversarial attacks: Fast Gradient Sign Method (FGSM) (I. Goodfellow 2014), Multi-step Projected Gradient Method (MS-PGD) (Aleksander Madry 2017), and Carlini-Wagner (CW) (Carlini and Wagner 2017b) attack.

**Fast Gradient Sign Method (FGSM)**   In this method (I. Goodfellow 2014), only one step gradient update is performed along the direction of the sign of gradient at each pixel. The adversarial perturbation is generated by taking the sign of the gradient of the loss function and multiplying it with $\epsilon$, the magnitude of the perturbation. These perturbations are formulated as:

$$\eta = \epsilon sign(\nabla_x J_\theta(x, l)) \tag{1}$$

**Basic Iterative Method (BIM)/Multi-step Projected Gradient Method (MS-PGD)**   In order to improve the FGSM attack, this method (Aleksander Madry 2017) was devised as an iterative version of the single-step FGSM attack for the l-$\infty$ adversary. The adversarial examples generated by these methods tend to be called "most-adversarial" examples as they are more aggressive and more likely to fool the classifiers. It can be formulated as following:

$$x^{t+1} = \prod_{x+S} (x^t + \epsilon * sgn(\nabla_x) L(\theta, x, y)) \tag{2}$$

**Carlini Wagner Attack** When certain adversarial defense techniques like Papernot's distillation approach (Papernot et al. 2016) started getting successful at preventing PGD and FGSM attacks, the Carlini Wagner (CW) (Carlini and Wagner 2017b) attack was introduced. The CW attack aims to solve the same problem as before, to find the minimally-distorted perturbation. However, CW attack uses a margin loss function $f(x, t)$ instead of a cross-entropy loss function $L(x, t)$ so that when the confidence is at a target value, $C(x') = t$ and margin loss is 0, $f(x, t) = 0$, the algorithm will then try to minimize the distance from $x'$ to $x$. The CW attack is said to be one of the strongest attacks ever developed having successfully beaten a lot of different defense techniques. It can be formulated as follows: where $f$ is defined as

$$f(x', t) = (max \, Z(x')_i - Z(x')_t)^+ \; such \, that \; i \neq t. \quad (3)$$

Minimizing $f(x', t)$ encourages the algorithm to find an $x'$ that has larger score for class t than any other label, so that the classifier will predict $x'$ as class $t$. Next, by applying a line search on constant $c$, we can find the $x'$ that has the least distance to $x$.

## Adversarial Defenses

As mentioned earlier there are broadly 2 categories of adversarial defense techniques, purification/rectification and adversarial detection. Purification/Rectification techniques can further be divided into sub-categories like adversarial retraining, pre-processing inputs, gradient masking/obfuscation (Papernot et al. 2016), etc. A lot of these adversarial purification/rectification defense techniques rely on a secondary deep learning architecture like neural networks, auto-encoders (AE) (Rumelhart, Hinton, and Williams 1986), and generative adversarial networks (GANs) (Goodfellow et al. 2014) to fulfill their objective. However, it has been shown that adversarial retraining and adversarial image purification techniques using AE like MagNet (Dongyu Meng 2017) and GANS (APE-GAN (S. Shen 2017), DefenseGAN (Samangouei, Kabkab, and Chellappa 2018)) end up lowering the overall classification accuracy (even for non-perturbed/clean images).

Adversarial detection based techniques, involve detection either using a secondary classifier (Grosse et al. 2017), by analyzing principal components (Hendrycks and Gimpel 2017; Li and Li 2017) of the input, or by comparing the distribution (Grosse et al. 2017; Feinman et al. 2017) of natural images to the distribution of adversarial examples. However, most of these defense techniques were also only validated on MNIST and CIFAR datasets and perform poorly when evaluated on higher resolution datasets (such as ImageNet). Additionally, these detection techniques tend to be successful only in a constrained setting with limited known attacks and datasets. They generally do not generalize well to newer attacks and a lot of these defense techniques have also shown to be ineffective against stronger attack types like CW. Some face recognition specific adversarial detection techniques have also been developed over the years, these include UAP (Agarwal et al. 2018), which uses pixel values and principal component analysis as features along

with a Support Vector Machine as a perturbation detection classifier. Another approach SmartBox (Goel et al. 2018), was a python based toolbox developed to evaluate adversarial robustness on a common face recognition benchmark, it also recommended a gaussian blur pre-processing based defense technique. Massoli et al. 2021 was another attempt to use neural networks with post-processing operations to detect adversarial samples. However these defense techniques were heavy-weight and couldn't be scaled for newer adversarial attacks easily.

## Metamorphic Testing Principles and Approach

Metamorphic testing(T. Y. Chen and Yiu 2020) uses tuples containing inputs and outputs where the "correctness" of a test case is defined by the "degree of change" in the output when applying a transformation to the input under consideration. A Metamorphic Relation is a conditional rule that defines the degree of the expected change in the output. We have previously shown that the metamorphic approach works for filtering adversarial examples achieving detection accuracy similar or better than deep learning approaches. The metamorphic approach is based on the assumption that an algorithm should be robust to minor transformations in the input, but the perturbation added to the adversarial sample could be rather unstable and would be exposed by the same transformation. To determine such metamorphic relations, primarily prior knowledge of the domain is used. We found that by quantifying the magnitude of change in the output data with a corresponding change to the input we could categorize output labels as this magnitude of change would be then used to classify data to relevant identities. Once these metamorphic relations are developed, they can be used as a reliable automated detection system for efficiently spotting anomalies in software systems without relying on extensive GPU computation. The general metamorphic approach for face recognition problem works as follows: Let $f(x) : X \rightarrow Y(X \subset R_n, Y \subset R_k)$ be a function that maps any given input face image, $X$ to a k-dimensional embedding using a pre-defined face embedding model. To generate an adversarial counterpart for the input $X$ being fed to the model, a minimal transformation of $\delta_x$ is assumed to be added. This perturbation then yields an *L2-distance* change of $\delta_y$ in the output embedding vector from embeddings corresponding to the original clean input embedding *Y*. Our hypothesis is based on the premise that since these added perturbations make an adversarial sample rather volatile, certain types of transformations could cause these adversarial input images to produce output distance changes $\delta_y$, which would be notably different from those produced by the transformations to clean face images. Since the trained face embedding model, itself is built to be invariant to a large number of perturbations in pose and illumination using a dataset of millions of images, the hypothesis should work for the transformations that we use. To create the metamorphic relations, we use statistical optimization methods to create effective decision thresholds that separate the behavior of transformations on clean and adversarial examples. These thresholds can therefore be used to predict whether a given input image is adversarial or not

with varying levels of confidence. Additionally, as compared to other deep learning-based approaches these metamorphic relations can be determined using a much smaller dataset and significantly less computational power thus making it much more practical to be used in real-time.

Our previous work (Mekala, Porter, and Lindvall 2020) shows that adversarial images on FaceNet behave very differently to ones from traditional white-box classification networks. We exploit this behaviour and employ an elegant technique of threshold optimization to create metamorphic relations. This technique focuses on maximizing the difference between true and false positive rate of classifications over the training data, effectively giving us the best possible thresholds for a given image transformation parameter. In this paper we build upon and validate it against multiple adversarial attacks and on various face embeddings models, we further also record the behavior of various combinations of transformations on clean and adversarial images.

## Implementation

### Face Embedding Setup

To set up a facial detection and recognition pipeline, we use Multi-task Cascaded Convolutional Networks (MTCNN) (Zhang et al. 2016) for the detection and alignment part. After a face has been detected, we crop out the face contour using bounding box information provided by MTCNN and normalize the face contour by subtracting all the pixels by 127.5, and then dividing them by 127.5 to map them to a [-1,1] range, these normalized images of face contours are then fed as input to a pre-trained Face Embedding (Florian Schroff 2015) architecture. The face embedding model is used to convert the input face images to high dimensional numerical representations (embeddings); which can then be used to numerically compare similarity between face images. We initially use FaceNet(Florian Schroff 2015) which uses Inception-ResNet (Christian Szegedy 2016) for the underlying network and is trained on the VGGFace2 (Cao et al. 2018) dataset. The FaceNet architecture trained maps every input face image to a unique embedding in the 512-dimensional space and achieves an accuracy of 99.25% on the Labeled Faces in the Wild (LFW) (Huang et al. 2007) dataset. We subsequently test the performance of our proposed algorithm on other face embedding models like LCNN (Wu et al. 2015), ArcFace (Deng, Guo, and Zafeiriou 2018) and CosFace (Wang et al. 2018) which were trained using the MS-Celeb-1M (Guo et al. 2016) dataset of 10 million images of around 100,000 individuals. The models achieve corresponding accuracies of 99.6%, 99.3% and 99.35% respectively on the Labeled Faces in the Wild (LFW) dataset. The LFW (Huang et al. 2007) dataset comprises over 13,000 images collected from the web, with each face being labeled as the name of the person and 1680 people having two or more distinct photos in the dataset. The face embedding modela used as part of the research conducted, were chosen based on variations in the underlying architectures or discriminative loss functions (Wang and Deng 2021). For example, we chose Light-CNN(Wu et al. 2015) due to it's specialized architecture, FaceNet(Florian Schroff

2015) due to it's Euclidean-distance-based loss function, CosFace(Wang et al. 2018) and ArcFace(Deng, Guo, and Zafeiriou 2018) due to their usage of angular/cosine-margin-based loss functions.

In order generate adversarial images for facial recognition models, we redefine the output of the facial embedding model from the embeddings of the input face image to instead a binary classification output based on similarity between a given image and a reference embedding as seen in Fig. 2. In Fig. 2 for each input image, we also provide a reference embedding. This reference embedding could either belong to the same person or some other person, the embedding is extracted and saved using the pretrained face embedding model. We modify the FaceNet flow graph such that it computes the embeddings from the input image and then calculates the cosine similarity between these input image embeddings and the reference embeddings. The cosine similarity between any two face image embeddings provides a measure of identity similarity between the people present in the two corresponding images. The output is given as [$CS$, $CD$], where $CS$ is the aforementioned cosine similarity and $CD$ is the cosine distance (or $1 - CS$). So, for an input image and corresponding reference embedding set, the output will be closer to $[1, 0]$ if the images belong to the same person and closer to $[0, 1]$ if they belong to different people.

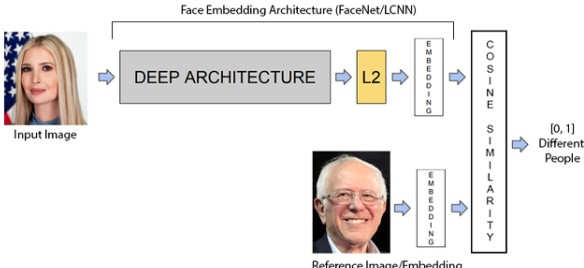

Figure 2: Binary Facial Recognition Problem

### Generation of Adversarial Examples

Adversarial examples are mathematically engineered noise perturbations to the input image that exploit the automatic differentiation capabilities of neural networks. The standard optimization technique of minimizing the loss function with respect to the input parameters across the gradient descent curve is exploited to instead maximize the loss with respect to input parameters. This enables the attacker to be able to find the least possible perturbation to any input image that maximises the computed loss, causing misclassification of a victim's true identity. We focus on 3 main attacks: Carlini and Wagner L_2 (CW) (Carlini and Wagner 2017b), Fast Gradient Sign Method (FGSM) (I. Goodfellow 2014) and Multi Step-Projected Gradient Descent (MS-PGD) (Aleksander Madry 2017). Using the LFW dataset, we create a dataset of 1200 adversarial images for each attack on the Modified FaceNet model. 600 of these are impersonation (falsely match to an impostor) attacks and other 600 are ob-

fuscation (falsely reject a genuine person) attacks. The set of 1200 adversarial images (per attack) achieve a model misclassification accuracy of 100% on all the face embedding models for all the attacks.

## Technology Stack

We use Tensorflow 2.0 on NVIDIA 1060 6GB for validating Face Embedding models and generating adversarial examples. The metamorphic defense implementation is done without GPU support to validate application in non-compute intensive environments.

## Proposed Metamorphic Threshold Detection Approach

As mentioned before, we attempt to create metamorphic relations using probabilistic thresholds of separation in the behavior of clean and adversarial examples when subjected to various forms of image transformations. We attempt to do so by utilizing the inherent robustness of face embedding models which would mean that clean images aren't affected that much i.e. are relatively stable when transformations are performed on the input image. However adversarial samples are generally created using the tight gaps in high dimensional spaces leading to attackers exploiting decision boundaries for target outputs. Thus they might be relatively unstable causing higher transformation-induced change to the output embedding from the model. Transforming a clean picture of Ivanka as seen in 3 causes a minimal change in the distance metric of the transformed image embedding from other image embeddings of the un-transformed image output. When an adversarial image of Chavez perturbed to a target label of Kofi Annan is subjected to the same transform, the un-transformed image distance metric shows extremely good proximity to Annan images as the output label, while the transformed variant's distance to embeddings of the same output label drops to 0.5. We use this information to hypothesize that for a given input image $x$, face detection model $Z$ and image transformation function $F$, the $sgn((Z(x) - Z(F(x))))$ value should be evidently more volatile and large for adversarial samples as compared to clean samples. The $sgn((Z(x) - Z(F(x))))$ distance is represented as a 3-dimensional vector $V$ comprising the 25, 50, and 75th percentile of distances between the embeddings of all the images before and undergoing the transformations. For a given clean image $X$ and its adversarial counterpart $Xa$, we could thus calculate the corresponding distance vectors $Vc$ and $Va$ respectively. The Frobenius-norm between the two vectors should give us the mathematical difference in output behavior of an adversarial image from a corresponding clean image when subjected to the same transformation. Once we have these difference vectors for each of the clean-adversarial example pairs, we can use a secondary classifier to develop threshold-based metamorphic relations to efficiently classify clean and adversarial samples generated using adversarial attacks. We have previously (Mekala, Porter, and Lindvall 2020) proposed and validated this on some non-linear transforms **cite** and PGD adversarial attack. For instance, when we use the minpool transformation, the L2-distance between transformation-led behavior of a clean and

| Attack Type | FGSM | PGD | CW | Comb |
|---|---|---|---|---|
| Blur | 71% | 67% | 70% | 68% |
| Horizontal Flip | 95% | 79% | 64% | 74% |
| Rotate | 55% | 69% | 52% | 54% |

Table 1: Adversarial Accuracy Detection for Linear Transformations

PGD adversarial counterpart being greater than 0.401282 allows us to classify it as an adversarial example with 93.64% confidence. For this paper, let $f$ be the embedding model function of the FaceNet face recognition pipeline and $V$ the transformation function used to obtain an optimum threshold. Additionally, let's define $Z$ as the classifier function that maps the input embedding to the mean distance metric of available identities and returns a tuple of classified identity label $Y$ and distance vector $D \subset R_3$. $D$ provides the 25th, 50th and 75th percentiles of distances from the input image to images of the identity $Y$ it is closest to for the untransformed image. As seen so far our approach requires no prior knowledge of the image being clean or adversarial and also no information about the target label of the input image is required.

## Experiment and Results

As mentioned in the previous section we use various classes of transformations to calculate the thresholds needed to efficiently detect adversarial examples. We broadly use 3 types of transformations: Affine, Non-Linear, and Combinations of Non-linear transformations.

## Affine Transformations

Affine transformations in specific are linear in nature, preserving co-linearity and distance ratios between points. The 3 transformation designs used below are partially drawn from our previous work (Rohan Reddy Mekala 2019). These designs were validated on white box attacks for smaller classification models trained on sparse data. As further enunciated through results in Table 1, we employ the following linear transforms to test their effectiveness towards detecting adversarial input samples on FaceNet in a black-box setting:

**Rotate Transform** : Rotate images to a randomly selected value between 180 and 200 degrees.

**Horizontal Flip Transform** : Mirror the images horizontally.

**Blur Transform** : Additive gaussian blur with $\sigma$ of 1.0.

We observe inconsistent adversarial image detection results with affine transformation methods, which can be attributed to the fact that models trained with data augmentations could conceal adversarial inputs when linear transformations are used. Blur and Rotate particularly seem to have very poor results. The results can be explained using Figure 4,which shows a kernel density plot of the differing behavior in distance metrics caused by these transformations. We can clearly see ambiguous separation in density curves for clean and adversarial examples caused by factors mentioned

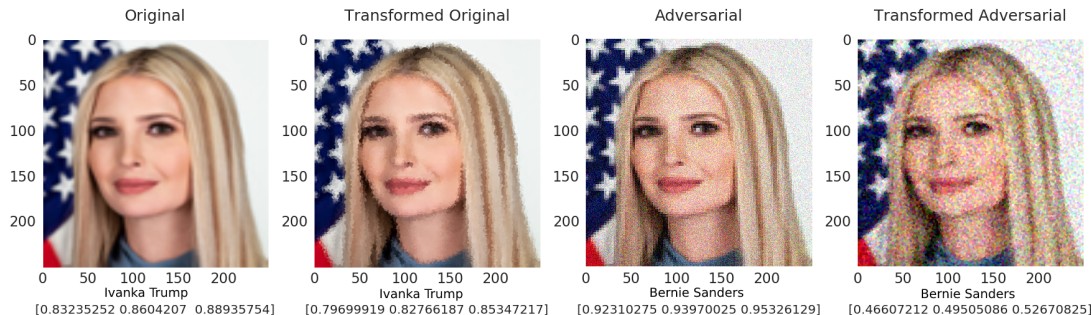

Figure 3: Effects of Non-linear Transformations on a Clean and Adversarial Image

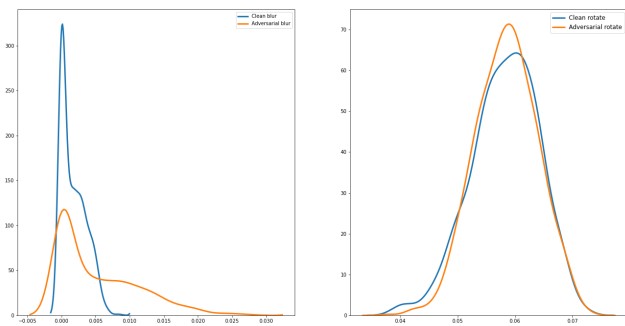

Figure 4: Density Plot for distance change in Clean and Adversarial (all attacks) examples for Affine Transformations

above; causing lackluster classification results. Horizontal Flip transformation has the relatively better results among affine transformations, achieving 95% for FGSM, 79% for PGD, 84% for CW and 84% for all attacks combined.

## Non-linear Transformations

Non-linear transformations involve using complex functions which do not preserve co-linearity and distance ratios between points, whereas random element transformations add pseudo random values generally from a distribution to the data-points. Following is a list of non-linear and random element transformation techniques used with a consolidated table of results achieved over test data in Table 2.

**Minpool Transform** Use kernel sizes of *4x4* to pool images channel-wise over a sliding window; taking the minimum pixel value per window. The resulting down-sampled image is then up-sampled using linear interpolation(H. Wu 2015).

**Maxpool Transform** Use kernel sizes of *4x4* to pool images channel-wise over a sliding window; taking the maximum pixel value per window. The resulting down-sampled image is then up-sampled using linear interpolation(H. Wu 2015).

**Medianpool Transform** Similar technique as above, except we take the median pixel value over a sliding window.

| Attack Type | FGSM | PGD | CW | Comb |
|---|---|---|---|---|
| Minpool | 96% | 94% | 92% | 93% |
| Maxpool | 95% | 92% | 91% | 92% |
| Medianpool | 96% | 98% | 94% | 96% |
| JPEG Compression | 84% | 85% | 89% | 86% |
| Gaussian Noise | 51% | 57% | 51% | 52% |
| MotionBlur | 86% | 83% | 83% | 84% |

Table 2: Adversarial Detection Accuracy for Non-Linear and Random Element Transformations

**JPEG Compression Transform** Decrease the quality of input images by a strength value of 20%.

**Gaussian Noise Transform** Add gaussian noise, sampled from a distribution of $N(0, 0.2 * 255)$.

**MotionBlur Transform** Blur images in a way that fakes camera or object movements, we use a kernel size of 15.

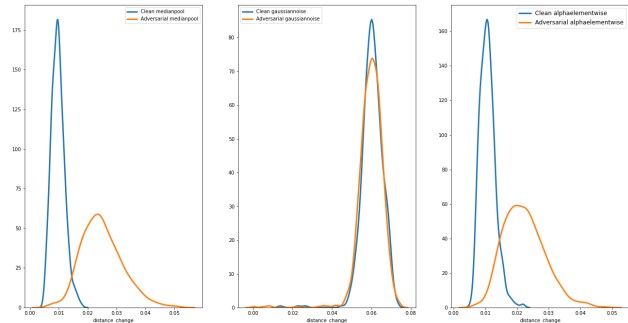

Figure 5: Density Plot for distance change in Clean and Adversarial (all attacks) examples for Non-Linear Transformations

We can see from results in Table 2 that while some random element transformations related to noise and snowflakes perform very poorly, other transformations have really good results. Transformation techniques involving Minpool, Maxpool, Medianpool and Alpha-Elementwise produce the best adversarial detection accuracies over 90%

| Attack Type | FGSM | PGD | CW | Comb |
|---|---|---|---|---|
| Min_JPEG | 96% | 98% | 93% | 97% |
| JPEG_Min_JPEG | 94% | 98% | 92% | 95% |
| Min_Median | 71% | 75% | 62% | 71% |

Table 3: Adversarial Detection Accuracy for Combination Transformations

| Defense Type | FGSM | PGD |
|---|---|---|
| UAP (Agarwal et al. 2018) | 74% | 61% |
| SmartBox (Goel et al. 2018) | 62% | 59% |
| Massoli et al. (Massoli et al. 2021) | 71% | 77% |
| Our Approach | 96% | 98% |

Table 4: Comparison with Other Adversarial Detection Implementation

| Defense Type | FGSM | PGD | CW | All |
|---|---|---|---|---|
| LCNN | 98% | 96% | 96% | 96% |
| ArcFace | 92% | 90% | 89% | 90% |
| CosFace | 91% | 91% | 90% | 91% |

Table 5: Adversarial Detection Implementation on other Face embedding methods (Best Results)

for all the attacks. As we can see from Fig. 5, the thresholds for seperation of clean and advresarial examples are well demarcated for transformations like medianpool and alpha-elementwise but ambiguous for transformations like gaussian noise addition, we can therefore conclude that random element transformations don't work well for adversarial sample detction. An additional observation to note is the high accuracy of the transformations even when all the adversarial attacks are combined together, which points to the technique being capable of adapting to new types of adversarial attacks presented in the future. Given the high accuracy achieved by non-linear transformations, we perform an additional experiment using combinations of some of our best non-linear transformation techniques and study their effects on the adversarial detection accuracy. We use the following non-linear transformation combinations and their results are shown in Table 3:

**Minpool - JPEG Compression Transform** Composite transform comprising 1) Implement Minpooling over a kernel size of 3x3 2)Decrease the quality of input images by a strength value of 20%.

**JPEG - Minpool - JPEG Transform** Composite transform comprising 1)Decrease quality of input images by a strength value of 10% 2) Implement Minpooling over a kernel size of 3x3 3)Repeat 1.

**Minpool - Medianpool Transform** Composite transform comprising 1) Implement Minpooling over a kernel size of 4x4 2)Implement Medianpooling over a kernel size of 4x4.

Table 3 illustrates that combinations of non-linear transforms gives much better results over many of the contributing individual transformations themselves. A combination of Minpooling and JPEG Compression transformation helps achieve our best adversarial detection accuracy of 97%. Based on the detection accuracy, the 2 best transformation techniques identified are 1)Minpool and JPEG compression combined(97%) and 2)Medianpool(96%).

**Comparison against other SOTA detection techniques**
Comparing our adversarial detection approach to current state-of-the-art techniques in adversarial detection in Table 4, our approach seems to be outperforming other approaches significantly. Our approach achieves higher adversarial detection accuracy for FGSM and PGD attacks compared to these other approaches but more significantly our approach also achieves a SOTA detection accuracy against CW attack and a dataset of combined attacks.

**Other Face Embedding Models**

We initially performed all our adversarial detection experiments on FaceNet. Once the best performing transforma-

tions were identified, we repeated the same experiments on the other aforementioned face embedding models (LCNN, ArcFace, and CosFace) to thoroughly validate the effectiveness of our approach. We can see in Table 5, that our approach achieves a detection accuracy of over 90% for all other face embedding methods. We do notice a small dip in accuracy for the arcface and cosface models and we plan on developing more transformations to possibly rectify and research this problem in future iterations.

## Conclusion and Future Work

Our metamorphic image transformation based defense pipeline delivers stellar results towards detecting different variations of adversarial attacks, specifically CW attack which is considered to be one of the strongest adversarial attacks (Athalye, Carlini, and Wagner 2018) in the world. We achieve a best-case adversarial detection accuracy of 96% for FGSM, 98% for PGD, 94% for CW and 97% for a dataset comprising of all adversarial attacks on FaceNet, while also achieving a detection accuracy of over 90% for other face embedding models. Our results are better than adversarial detection approaches considered state-of-the-art at present 4. Our detection approach also does well when a dataset of all the adversarial attacks is considered making it adversarial attack agnostic while detecting adversarial samples. Our approach requires no knowledge of the input data or architecture/parameters of the model under attack. Additionally, our adversarial detection process is not computationally expensive and does not require the true embedding distance/classification output of a face. Based on research available, we are the first to set a benchmark for defense against multiple state-of-the-art adversarial attacks using metamorphic approach in the face recognition domain. We hope our work will continue to inspire more research towards using metamorphic principles for defense against adversarial attacks.

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
