# OpenReview forum: "Metamorphic Adversarial Detection Pipeline for Face Recognition Systems"
_AAAI.org/2022/Workshop/AdvML — AAAI-22 AdvML Workshop LongPaper_

### Official Review · Reviewer_Kysc · 2021-11-29
**Metamorphic Adversarial Detection Pipeline**

**Rating:** 6
**Confidence:** 4

**Review:**

This paper proposes to use statistical techniques to learn optimal thresholds of separation between clean and adversarial examples, and achieves excellent detection accuracies.

Experimental performance can be demonstrated that the method can effectively detect adversarial examples and achieve a good performance. However, the weakness is also listed as follows:

1.	More datasets should be introduced for comprehensive comparisons, such as MS-Celeb-1M[1].

2.	Will different face detectors affect the final performance?

[1] MS-Celeb-1M: A Dataset and Benchmark for Large-Scale Face Recognition

---

### Official Review · Reviewer_uzxp · 2021-11-30
**Review for a metamorphic adversarial detection**

**Rating:** 6
**Confidence:** 4

**Review:**

This paper propose a metomorphic adversarial detection method for face recognition system. The idea is clear and writing is good. I think it is a interesting topic. There is some advice. 1)Some pictures in this paper seem similar to the previous paper. I strongly suggest a replacement. 2)Face recognition systems involve face alignment before the verification. It is not clear wheather transformations like rotation remain effective.

---

### Decision · Program_Chairs · 2021-12-01

**Decision:**

Accept (Long Paper)

**Comment:**

Both reviewers agree to accept this paper. The authors are encourages to address the reviewers' comments in the camera-ready version.